# Real Space and Time Imaging of Collective Headgroup Dipole Motions in Zwitterionic Lipid Bilayers

**DOI:** 10.3390/membranes13040442

**Published:** 2023-04-18

**Authors:** Dima Bolmatov, C. Patrick Collier, Dmitry Zav’yalov, Takeshi Egami, John Katsaras

**Affiliations:** 1Department of Physics and Astronomy, University of Tennessee, Knoxville, TN 37996, USA; 2Shull-Wollan Center, Oak Ridge National Laboratory, Oak Ridge, TN 37831, USA; 3Center for Nanophase Materials Sciences, Oak Ridge National Laboratory, Oak Ridge, TN 37831, USA; 4Department of Physics, Volgograd State Technical University, Volgograd 400005, Russia; 5Department of Materials Science and Engineering, The University of Tennessee, Knoxville, TN 37916, USA; 6Materials Science and Technology Division, Oak Ridge National Laboratory, Oak Ridge, TN 37831, USA; 7Sample Environment Group, Neutron Scattering Division, Oak Ridge National Laboratory, Oak Ridge, TN 37831, USA

**Keywords:** zwitterionic lipid bilayers, POPC, DPPC, headgroup dipoles, collective dipole motions, Van Hove correlation functions, heat transfer, surface undulations, piezoelectricity, flexoelectricity

## Abstract

Lipid bilayers are supramolecular structures responsible for a range of processes, such as transmembrane transport of ions and solutes, and sorting and replication of genetic materials, to name just a few. Some of these processes are transient and currently, cannot be visualized in real space and time. Here, we developed an approach using 1D, 2D, and 3D Van Hove correlation functions to image collective headgroup dipole motions in zwitterionic phospholipid bilayers. We show that both 2D and 3D spatiotemporal images of headgroup dipoles are consistent with commonly understood dynamic features of fluids. However, analysis of the 1D Van Hove function reveals lateral transient and re-emergent collective dynamics of the headgroup dipoles—occurring at picosecond time scales—that transmit and dissipate heat at longer times, due to relaxation processes. At the same time, the headgroup dipoles also generate membrane surface undulations due a collective tilting of the headgroup dipoles. A continuous intensity band of headgroup dipole spatiotemporal correlations—at nanometer length and nanosecond time scales—indicates that dipoles undergo stretching and squeezing elastic deformations. Importantly, the above mentioned intrinsic headgroup dipole motions can be externally stimulated at GHz-frequency scale, enhancing their flexoelectric and piezoelectric capabilities (i.e., increased conversion efficiency of mechanical energy into electric energy). In conclusion, we discuss how lipid membranes can provide molecular-level insights about biological learning and memory, and as platforms for the development of the next generation of neuromorphic computers.

## 1. Introduction

The study of collective molecular motions in hydrated phospholipid bilayers is a challenging area of science [1,2,3]. Although almost all liquid crystalline lipid bilayers undergo large molecular displacements, their intermolecular interactions are lipid specific [4,5,6,7]. In biological membranes, lipids are in constant motion—due to thermal energy—and their movements impact their physical properties, such as bilayer viscosity, energy and heat transfer, lipid diffusion, and overall morphology [8,9,10,11,12]. Importantly, lipid bilayers are the underlying structures of biological membranes and define the cell’s extracellular and intracellular environments [13,14], and enable a range of biological processes [15].

Compared to other lipid bilayer dynamics, collective molecular motions, at their fundamental time and length scales, have been less studied experimentally [16,17,18,19,20]. In contrast, there have been considerable theoretical developments and analytical models that have focused on understanding membrane collective motions [21,22,23,24]. However, almost without exception, these developments have been implemented in reciprocal space, making them somewhat cumbersome. On the other hand, demand for the development of theoretical and analytical frameworks using real spatial and temporal analyses at fundamental length and time scales, has been growing.

Zwitterionic lipid bilayers (see Figure 1) contain both positive (choline, blue) and negative (phosphate, orange) functional groups, resulting in a relatively high dipole moment at physiological pH [25,26,27,28]. Phospholipids such as 1-palmitoyl-2-oleoyl-sn-glycero-3-phosphocholine (POPC, 16:0/18:1 PC) and 1,2-dipalmitoyl-sn-glycero-3-phosphocholine (DPPC, 16:0/16:0 PC) spontaneously self assemble into bilayers in the presence of water (Figure 1), with their hydrophobic acyl tails forming the bilayer’s hydrophobic core, while their headgroups interact with the aqueous phase [29]. Both lipids occur naturally, and are involved in different cellular processes on length scales ranging from the nanometer to the micrometer [30,31,32,33,34].

Here, we developed an approach using the Van Hove correlation function in one, two, and three dimensions (1D, 2D, and 3D, respectively) to study the spatial and temporal correlations of zwitterionic PC headgroup dipoles. X-ray and neutron scattering techniques such as, diffraction have extensively been used to study the structural organization of lipids in biomembranes [38,39,40], including spatial lipid-lipid pair distribution functions and snapshot-like time evolution of their spatial correlations. Although useful, such information alone is insufficient for the understanding of collective dynamics and relaxation processes, which contain both spatial and temporal correlations.

We show that in 2D and 3D, the spatiotemporal images of collective motions are consistent with experimentally observed, common dynamic features associated with fluids. However, in 1D, analysis of the Van Hove function revealed lateral transient and re-emergent picosecond dynamics of the headgroup dipoles that transmit and subsequently dissipate heat. Importantly, the PC headgroup dipoles also generate slower membrane surface undulations—due to collective dipole tilting—and can undergo both stretching and squeezing elastic deformations. In theory, the headgroup dipole motions can also be externally stimulated at GHz-frequency scale, enhancing their piezoelectric capabilities (i.e., increased conversion efficiency of mechanical energy into electric energy).

## 2. Materials and Methods

### 2.1. Coarse-Grained Molecular Dynamics (MD) Simulations

We performed MD simulations using GROMACS (http://www.gromacs.org, accessed on 4 July 2022) and the MARTINI 2.2p force field [41]. A square model membrane with periodic boundary conditions was built using the CHARMM-GUI [42,43]. The number of lipids in each bilayer leaflet was 512, with a 22 Å water layer thickness on either side of the bilayer membrane. Pressure was set to 1 bar at 37 ${}^{\circ }$C. The MD procedure was performed using the following three steps: (1) During the first simulation stage, the system energy was minimized by minimizing the intermolecular forces; (2) The system was brought into thermodynamic equilibrium using the NVT canonical ensemble, where T is the absolute temperature, N is the number of atoms, and V is the volume. The Thermostat V-Rescale with a time-constant of 1 ps was used for all the simulations, and the groups of atoms belonging to water and lipids were thermalized separately. Thermalization took place over 30 ns in steps of 20 fs; and (3) During the production stage, the system was in the NPT ensemble using the V-Rescale thermostat with a coupling time constant of 12 ps for the barostat. The simulation was carried out for 6 ns, in 20 fs increments. For the second and third simulation steps, van der Waals interactions were cut off at 1.1 nm and the electrostatic interactions, using the PME algorithm, were cut off at 1.2 nm.

### 2.2. 3D, 2D, 1D Van Hove Correlation Functions

The Van Hove correlation function describes the probability density of finding a particle *A* at a distance *r* and time *t*, when a particle *B* is at time t=0 [44,45]. Previously, Van Hove function analysis has been used to analyze ultrafast real space dynamics of liquids using inelastic X-ray scattering [46,47,48,49]. Below, we define the Van Hove correlation function in 1D, 2D, and 3D to study the spatiotemporal collective motions of PC headgroup dipoles associated with POPC and DPPC lipid bilayers.

The Van Hove correlation function has two independent terms, i.e., a self part that describes diffusion properties of a particle in a local neighborhood, and a distinct part that describes collective properties of a particle correlated with neighboring particles [45]. Below, we define Van Hove correlation functions which contain both self and distinct parts.

The 3D Van Hove correlation function, $G^{3D}\left(r,t\right)$, which describes the distribution of *A* particles relative to *B* particles in a given volume, can be written as follows:(1)$$G_{AB}^{3D}\left(r,t\right)=\frac{V}{4\pi r^{2}N_{A}N_{B}}\sum_{i,j}\delta (r-|r_{i}^{A}\left(t\right)-r_{j}^{B}\left(0\right){|}_{3D}),$$
where $|r_{i}^{A}\left(t\right)-r_{j}^{B}{\left(0\right)|}_{3D}$ is the distance between particles *i* and *j*, with types *A* and *B*, respectively. $N_{A}\left(N_{B}\right)$ is the number of A(B) particles, *t* is the time, and *V* is the volume. Analogously, the 2D Van Hove correlation function $G^{2D}\left(r,t\right)$ can be written as:(2)$$G_{AB}^{2D}\left(r,t\right)=\frac{S}{2\pi rN_{A}N_{B}}\sum_{i,j}\delta (r-|r_{i}^{A}\left(t\right)-r_{j}^{B}\left(0\right){|}_{2D}),$$$|r_{i}^{A}\left(t\right)-r_{j}^{B}{\left(0\right)|}_{2D}$ is the distance between particles *i* and *j* with particle types *A* and *B*, respectively, in a given plane where *S* is the area of the system.

Finally, we define the 1D Van Hove correlation function $G^{1D}\left(r,t\right)$ as:(3)$$G_{AB}^{1D}\left(r,t\right)=\frac{L}{N_{A}N_{B}}\sum_{i,j}\delta (r-|r_{i}^{A}\left(t\right)-r_{j}^{B}\left(0\right){|}_{1D}),$$
where $|r_{i}^{A}\left(t\right)-r_{j}^{B}{\left(0\right)|}_{1D}$ is the distance between the *i* and *j* particles with particle types *A* and *B*, respectively, along a given axis, where *L* is the length of the system along the axis.

For all Van Hove correlation functions, the headgroup bilayer (Figure 1) position, $r_{i}^{A}$, is the center of mass of the *i*th choline (NC3-bead), and position $r_{i}^{B}$ is the center of mass of the phosphate (PO4-bead) group. Thereafter, we will use the following notations: $G_{NP}^{1D}\left(r,t\right)$, $G_{NP}^{2D}\left(r,t\right)$, and $G_{NP}^{3D}\left(r,t\right)$. We would like to clarify that we used the phosphate bead as the r(0) of the Van Hove function and not the choline bead. We should also point out, that the difference between the choline and the phosphate bead r(0) is minimal, as such, we did not elaborate further on this point in the manuscript.

## 3. Results

### 3.1. Spatiotemporal Imaging of Collective Headgroup Dipole Motions and Relaxation Processes in Hydrated Phospholipid Bilayers

POPC and DPPC bilayers differ only in the chemistry of their acyl chains namely, DPPC has two 16:0 chains, while POPC has a 16:0 chain at its sn-1 position and an 18:1 at its sn-2 position. However, the presence of a double bond in the case of POPC (Figure 1), causes its physicochemical properties to be different from those of DPPC. As a result, compared to DPPC, POPC bilayers are more disordered and have a much lower gel-to-liquid-crystalline phase transition temperature (41 ${}^{\circ }$C for DPPC vs. −2 ${}^{\circ }$C for POPC)—at approximately 37 ${}^{\circ }$C, DPPC bilayers undergo a so-called gel to ripple phase transition [50].

The Van Hove function captures spatiotemporal correlations in the form of particle density fluctuations and as such, can be used to directly access density fluctuations measured via inelastic neutron scattering (INS) [51]. Figure 2 displays 2D and 3D Van Hove correlation functions (see Equations (Equation 1) and (Equation 2)) of POPC (a,b), and DPPC (c,d) headgroup dipoles, whose ultrafast dynamic behavior is consistent with what has been previously observed in fluids by IXS [46,47,48,49] and INS [51]. For both POPC and DPPC bilayers, headgroup dipoles have sharp intensity peaks located at ∼0.45 nm, consistent with nearest neighbor chain-chain distances observed in PC bilayers (∼0.4 nm). The red sharp peaks (in the normalized intensity scale bars located to the right of Figure 2) indicate that the PC headgroups experience rattling or “cage-like” dynamics in the first 2–4 ps, followed by a long lasting relaxation process, i.e., the intensity drops down from “red” to “white”. Interestingly, $G_{NP}^{2D}\left(r,t\right)$ correlation functions are more informative than $G_{NP}^{3D}\left(r,t\right)$ (see Figure 2), i.e, Figure a,c vs. Figure b,d. Specifically, the “caging phase” of the collective dynamics associated with the headgroup dipoles undergoes a clear transition (white arcs around the red peaks), and the traces of the relaxation phase are more detailed and last significantly longer. As such, it is physically justified to independently analyze the behavior of the $G_{NP}^{1D}\left(r,t\right)$ correlation function of the POPC and DPPC headgroup dipoles, both laterally and along the *z*-axis.

Figure 3 shows 1D Van Hove correlation functions (see Equation (Equation 3)) of POPC (a,b) and DPPC (c,d) headgroup dipoles undergoing lateral motion. Interestingly, analysis of the Van Hove correlation function in reduced dimensions provides additional insights, which are consistent with previous IXS results from oriented lipid bilayers [52]. Specifically, 1D analysis of lateral headgroup dipole motions reveals spatiotemporal correlations that are buried in higher dimensions. It is worth noting that $G_{NP}^{1D}\left(r,t\right)$ correlation functions are identical along any direction within the bilayer plane (Figure 1), since both POPC and DPPC membranes are isotropically disordered in the lateral direction, as well as along the “x” and “y” axes. Furthermore, over short times (0–25 ps), dipoles in the first coordination shell experience the strongest caging events (Figure 3a,c), slowly relaxing at longer times with well-defined attenuating traces. During the caging phase, lipids are capable of supporting both longitudinal collective motions, in the form of compression waves, and transverse collective displacements, in the form of shear waves. This means that mechanical stress is localized, dynamically squeezing both the lipids and the bound interfacial water molecules, thereby making the bilayer elastic [53,54]. At longer times, transverse restoring forces become negligible, i.e., undergoing the elastic-to-viscous crossover [53], and lipid molecules are no longer responsive to shear waves, resulting in a viscous local lipid environment. Therefore, the localized mechanical stress is forced to propagate forward due to continuous impact from compression waves. Of note, these observations are consistent with previously reported reciprocal space data from IXS [54].

Of note, unlike in simple liquids [55,56], the mechanical stress in phospholipid membranes propagates laterally, bouncing back and forth between neighboring local lipid environments (Figure 3a,c). Extending this in space and time (Figure 3b,d) reveals a mesh-like vibrational pattern, reminiscent of the surface of a “boiling” viscous liquid [54]. This spatiotemporal image—coexistence of high intensity (compressed) and low intensity (diffused) regions—also lends further evidence supporting the mechanism for passive transmembrane transport of small molecules and solutes revealed by IXS experiments [52]. Due to increased disorder, POPC lateral correlations are also less intense than those of DPPC bilayers, as highlighted by the randomly-circled red intensity peaks in Figure 3 [(b) vs. (d)] and Figure 4 [(a) vs. (b)]. Furthermore, Figure 4 shows a periodic time evolution of the spatiotemporal correlation intensity peak within the first coordination shell generated by POPC (a) and DPPC (b) lipid headgroups, respectively. The insets to the figure, generated using Fourier transforms, show power spectra of POPC and DPPC headgroup dipoles ranging between ∼1 GHz and ∼30 GHz. Importantly, the wiggling patterns of the collective lateral headgroup dipole motions—represented in the dark blue low intensity areas—are clearly featured. For instance, arising at ∼400 ps along r= 0.8 nm, potentially coupling to bilayer surface undulations existing at slower times and frequencies.

Figure 5 shows 1D Van Hove correlation functions (see Equation (Equation 3)) of POPC (a,b) and DPPC (c,d) headgroup dipoles, characterizing their collective motions along the *z*-axis, i.e., $G_{NP}^{1D}\left(r,t\right)$ (Figure 1). In contrast to their lateral spatiotemporal correlations (Figure 3), the red intensity band is continuously filled (0–0.15 nm), the result of continuous stretching and squeezing elastic deformations of the headgroup dipoles (Figure 5). Moreover, out-of-plane dipole motions evolve continuously in time (see red intensity band), where the highest intensity peak fluctuates at around 0.1 nm (see red line). The white intensity line located at ∼0.16 nm defines the transient edge phase of headroup dipole correlations, with a smooth oscillatory behavior taking place between ∼0.12 nm and ∼0.18 nm—no correlations are observed beyond 0.2 nm. Interestingly, both intensity lines (red and white) evolve coherently in time, implying that headgroup dipoles experience collective tilting, which generate surface waves or lipid bilayer surface undulations [58,59,60].

Figure 6 shows the time evolution of the $G_{NP}^{1D}\left(r,t\right)$ first peak height of POPC (a) and DPPC (b) lipid headgroup dipoles. The insets to the figure generated using Fourier transforms, show the power spectra of the POPC and DPPC headgroup dipoles ranging between ∼0.05 GHz and ∼0.5 GHz. Such continuous collective movements of lipids in a bilayer are known as membrane surface undulations and play important roles in different biological processes [61,62,63].

### 3.2. Dipole-Dipole Interactions in Lipid Bilayers

We estimated the individual vibrational frequency of neighboring lipids due to their dipole-dipole interactions along the *z*-axis. Specifically, we considered two lipid molecules from the inner and outer bilayer leaflets oscillating along the *z*-axis. The oscillating frequency of the pair of PC headgroup dipoles can be estimated from the following expression:(4)$$m\frac{d^{2}z}{dt^{2}}=-\frac{dV}{dz},$$
where m= 760 and 734 are the masses of the POPC and DPPC molecules, respectively, and *V* is the dipole-dipole interaction potential. The first order derivative (right term in Equation (Equation 4)) is taken at the average distance between the dipoles. The potential produces a restoring force *F* that tends to relax at the dipoles’ rest positions. The force *F* can then be estimated as
(5)$$F=-\frac{dV}{dz}∼-\frac{p^{2}}{4\pi \epsilon_{0}\epsilon z^{4}},$$
where p=ql is the dipole moment of a headgroup, *q* is the elementary charge, l≈ 0.55 nm is the average distance between the choline and phosphate (N−P) beads (Figure 1), and ϵ ($\epsilon_{0}$) is the dielectric constant (vacuum permittivity). The angular frequency ω of dipole-dipole oscillations can be written as:(6)$$\omega =\sqrt{\frac{p^{2}}{4\pi \epsilon_{0}\epsilon mr^{5}}},$$
where r≈4.5 nm is the average distance between the dipoles. Therefore, $\omega \approx 2\;\times \;10^{-9}$ s${}^{-1}$ and subsequently its vibrational frequency $f=\frac{2\pi }{\omega }=0.32$ GHz. Finally, this estimated value (0.32 GHz) of the dipole-dipole vibrational frequency is within the frequency range of both POPC and DPPC headgroup dipoles (see insets to Figure 6) calculated using MD simulations, implying that such dipole-dipole interactions contribute to the generation of lipid bilayer surface undulations.

## 4. Discussion and Conclusions

Biological membranes have recently been shown that they may be good systems for understanding the processing efficiency of the human brain [64]. In this regard, lipid membranes can also be utilized as neuromorphic platforms (e.g., artificial neural networks and memory computing crossbar architectures) [65,66,67]. For example, in neuroscience, long-term potentiation (LTP) is a long-lasting form of synaptic plasticity leading to formation, compartmentalization, and consolidation of biological memory [68]. A recent study showed that PC bilayers enable the formation of LTP mimicking the hippocampal LTP [69], previously observed in mammals and birds [70]. These memory effects can be stimulated in phospholipid membranes in response to external stimuli in the form of cyclic electric fields [57]. In this regard, piezoelectricity—a conversion of mechanical energy of lipids into electric energy, may play a major role underpinning the molecular mechanisms of memcapacitive energy storage that results in the formation of LTP [69]. Indeed, phospholipid bilayers are capable of supporting such flexoelectric and piezoelectric effects, which are stimulated by mechanical (compression and shearing), electric, optical, and magnetic stimuli, inducing the squeezing, stretching, and tilting of lipid molecules [71,72,73,74,75]. In this regard, the knowledge about intrinsically collective lipid motions may help us to better understand the flexoelectric and piezoelectric properties of lipid bilayers in the presence of external stimuli [76,77,78,79,80,81,82,83,84].

In this manuscript, we have provided spatiotemporal imaging of zwitterionic lipid bilayer collective motions, which can potentially be stimulated by external stimuli, by matching the bilayer’s vibrational frequencies, thus enhancing its piezoelectric effects [57]. Specifically, we introduced 1D, 2D, and 3D Van Hove correlation functions to visualize intrinsic headgroup dipole motions. 2D and 3D spatiotemporal imaging was shown to be consistent with common vibrational features observed in liquid systems. 1D analysis was then applied to study lateral and out-of-plane dipole motions, revealing their unique vibrational patterns. Lateral collective motions of headgroup dipoles experience ultrafast—2–4 ps—rattling dynamics, followed by long-lasting relaxation events (up to 25 ps). The extended space and time imaging of such motions showed that mechanical stress is capable of propagating back and forth, bouncing between neighboring lipid dipoles as a function of time. Mechanical stress exists in the form of localized compression and shear waves. The subsequent relaxation events enabled the release of mechanical stress, propagating it forward (both spatially and temporally), and dissipating it in the form of heat at longer times. Remarkably, the same headgroup dipoles also generated collective lipid bilayer surface undulations at ns time scales, as revealed by our 1D out-of-plane analysis. In contrast to lateral ps transient caging dynamics, out-of-plane collective dipole tilting generated continuous long-lasting—ns time scale—membrane surface undulations.

## Figures and Tables

**Figure 1 membranes-13-00442-f001:**
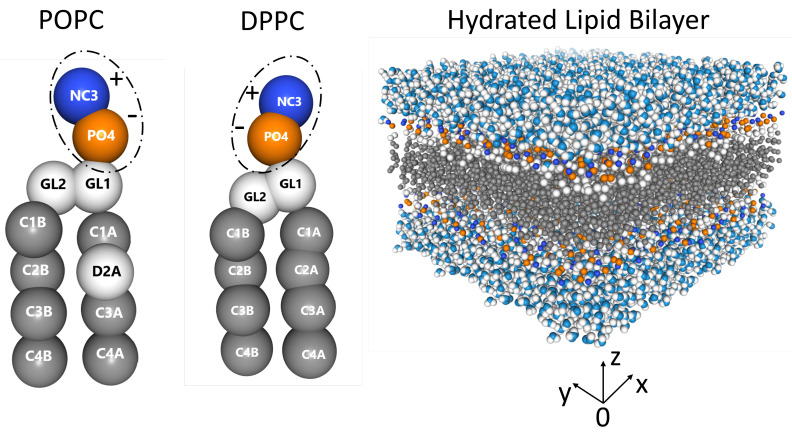
From left to right: POPC (16:0/18:1 PC), DPPC (16:0/16:0 PC), and a hydrated PC bilayer. Dashed ellipses around the POPC and DPPC phospholipid headgroups highlighting their positive charged choline (blue) and negative charged phosphate (orange) groups. The lipid bilayer is fully hydrated with the hydrophilic PC headgroups interacting with polar water molecules, which continuously experience the formation and breakup of hydrogen bonds at picosecond time scales [35,36,37].

**Figure 2 membranes-13-00442-f002:**
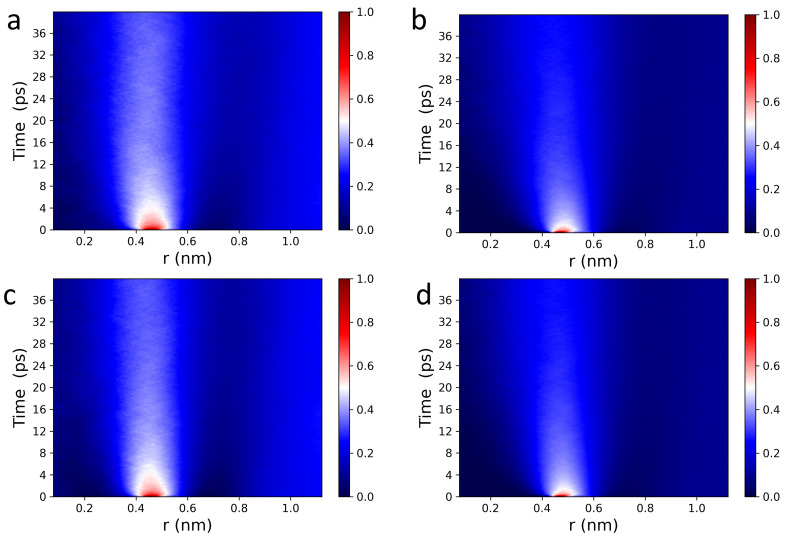
2D and 3D Van Hove correlation functions (see Equations (Equation 2) and (Equation 3)) of POPC (**a**,**b**) and DPPC (**c**,**d**) PC headgroup dipoles consist of choline-phosphate functional groups (see Figure 1). (**a**,**c**) $G_{NP}^{2D}\left(r,t\right)$ and (**b**,**d**) $G_{NP}^{3D}\left(r,t\right)$ correlation functions of PC headgroups at 37 ${}^{\circ }$C contain all of the major spectral features of liquid systems measured using IXS [46,47,48,49] and INS [51].

**Figure 3 membranes-13-00442-f003:**
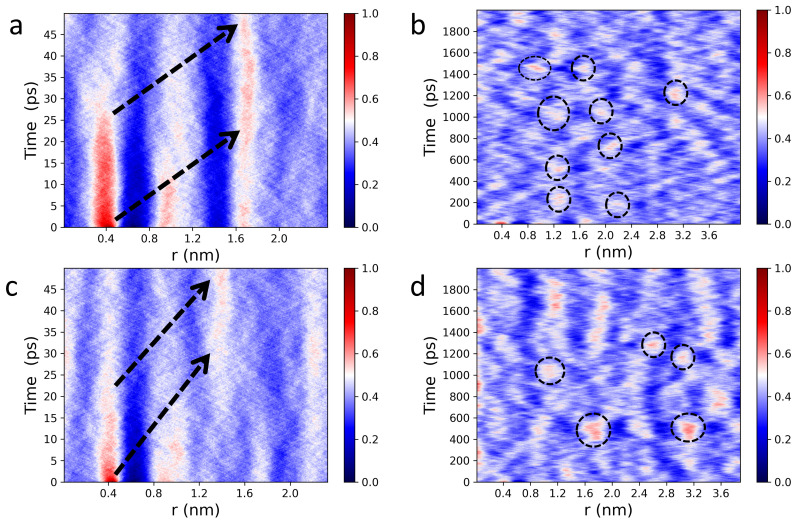
Lateral, spatiotemporal pair correlations for POPC (**a**,**b**) and DPPC (**c**,**d**) headgroup dipoles calculated from Equation (Equation 1). Wide, red line intensity peaks (**a**,**c**) indicate strongly correlated “cage-like” dynamics, followed by smooth relaxation processes (nearly straight white traces) along the first, second, and third coordination shells. Black arrows indicate the propagation of membrane lateral stress, in real space and time, migrating through the different coordination shells. An extended spatiotemporal picture of collective headgroup dipole motions (**b**,**d**). The circled intensity peaks highlight areas of localized mechanical stress within otherwise relaxed neighborhoods in space and time. “Wiggling” low intensity areas (dark blue regions) feature the co-existence of different collective vibrational patterns at the water-bilayer interface.

**Figure 4 membranes-13-00442-f004:**
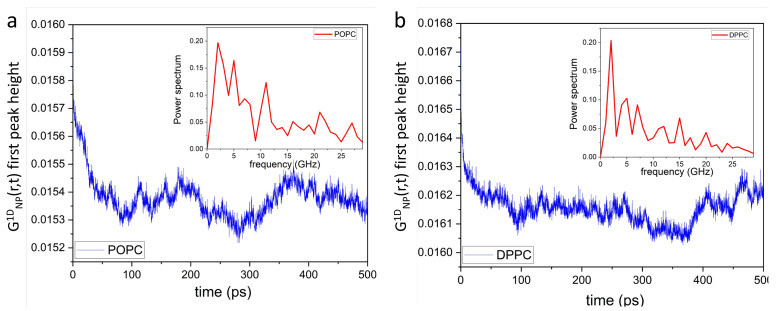
Time evolution of the $G_{NP}^{1D}\left(r,t\right)$ first peak height along r≈ 0.8 nm of lateral POPC (**a**) and DPPC (**b**) headgroup dipole spatiotemporal correlations. Insets show their respective power spectra, where their frequency bands range between ∼2 GHz and ∼25 GHz. Importantly, phospholipid bilayers can be externally stimulated within these frequency bands through dynamic impedance spectroscopy [57], increasing conversion efficiency of mechanical energy into electric energy.

**Figure 5 membranes-13-00442-f005:**
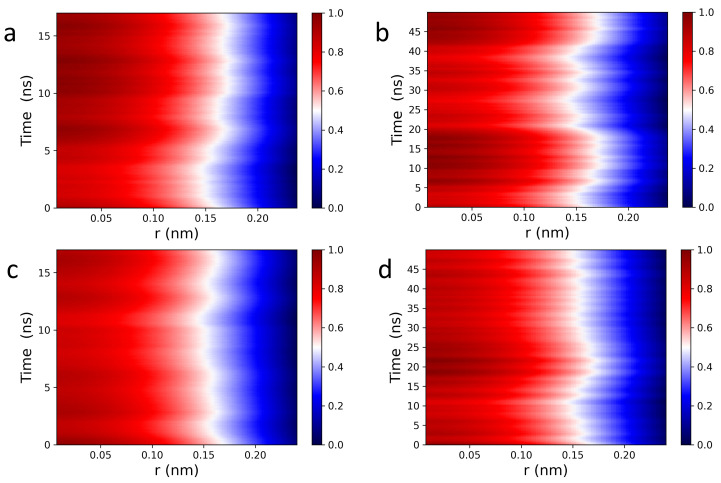
Spatiotemporal images of POPC (**a**,**b**) and DPPC (**c**,**d**) headgroup dipoles (along the *z* axis, see Figure 1) correlations calculated from Equation (Equation 1). Headgroup dipoles experience squeezing and stretching motions, filling the red intensity band from 0 nm up to 0.15 nm. The continuous, in time, oscillatory edge phase behavior of $G_{NP}^{1D}\left(r,t\right)$, implies the existence of lipid bilayer surface undulations generated by the collective tilting of headgroup dipoles.

**Figure 6 membranes-13-00442-f006:**
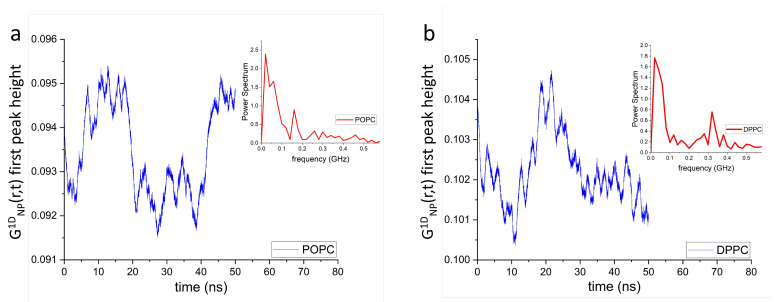
Time evolution of the $G_{NP}^{1D}\left(r,t\right)$ first peak height of POPC (**a**) and DPPC (**b**) headgroup dipole correlations along the *z*-axis. Insets show the power spectra of the $G_{NP}^{1D}\left(r,t\right)$ first peak height, where their frequency bands range between ∼0.05 GHz and ∼0.5 GHz. It is worth noting that lipid bilayers can be stimulated at these frequencies using dynamic impedance spectroscopy [57]. Doing so, may help to increase the conversion efficiency of mechanical energy into electric energy.

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
