# Peer review of "Real Space and Time Imaging of Collective Headgroup Dipole Motions in Zwitterionic Lipid Bilayers"

_membranes, 2023, doi:10.3390/membranes13040442_

Round 1

Reviewer 1 Report

In the article entitled 'Real space and time imaging of collective headgroup dipole motions in zwitterionic lipid bilayers' authors are using van Hove approach (1D, 2D and 3D) to analyze probability density of finding dipole partilces in lipid membrane systems simulated with coarse-grain MD. By setting the temperature at 37 degrees, they investigate lipid membranes in two state - liquid (POPC) and gel-rippled (DPPC).
The article is written well and data is presented clearly. Nevertheless, I have a few remarks.

1) When authors are using Van Hove analysis they calculate the distance between each phosphate and choline bead pairs. As far as I understand the phoshpate is the one that is taken in calculations as r(0). Did authors check if choline bead is the one treated as r(0) the van Hove function would change?

2) How do authors deal with PBC in their analysis? Molecules jumping over PBC could be a serious concern. For instance in lateral diffusion studies this was emphesized as an important factor (see J. Chem. Phys. 153, 024116 (2020); https://doi.org/10.1063/5.0008312). I'm guessing that in this study sudden jumps of beads to other site of system box would contribute to error in Van Hove function.

3) For analysis in 2D and 3D are authors including the movement of the system? The membrane system itself is also moving - the center of mass position would differ in time. Shouldn't this be included for r(0) to avoid introduction of error due to movement of whole system?

4) I'm a little puzzled by lack of difference between DPPC and POPC systems. Since both of those lipids do not differ significantly in coarse-grain representation, is it possible that lack of difference between those two systems comes from limitation of this representation? I'll again back this up with diffusion coefficient studies, as they are also based on movement of molecules - for instance in doi.org/10.1080/08927022.2013.840902 diffusion coefficient was 4-times higher than in atomistic simulations. Are authors sure that the use of coarse-grain MD for this type of analysis does not lead to severe loss of information?

5) I don't understand why authors are introducing frequency representation in Figure 4. It is barely discussed and doesn't seem to be relevant to the topic of the article at all.

6) Finally, are authors sure that the effect they report are due to dipole properties of lipid molecules? Lipid membranes are dynamic entity to begin with, and reported G1D such as in Fig 4 and Fig6 hardly differ from movement of any atom in the system. Fluctuations of lipid membranes, observed in time regime of tens of ns, are often used to estimate mechanical properties of lipid membranes. I think authors should emphesize more on the advantages of analyzing dipoles and the benefits of that compared to other studies. After all mechnical stress is also determined based on orientation of lipid molecules, dynamics of simulation box sizes, dynamics of area per lipid and/or membrane thickness, etc.
I also think authors should write more what additional and advantageous information does collective molecular motion provide rather than only stating that it has been less studied. Finally, they should emphesize what additional information the collective molecular motion did provide in investigated case.

Author Response

Referee #1: The article is written well, and data is presented clearly.

Comments:

Referee: 1) When authors are using Van Hove analysis they calculate the distance between each phosphate and choline bead pairs. As far as I understand the phoshpate is the one that is taken in calculations as r(0). Did authors check if choline bead is the one treated as r(0) the van Hove function would change?

Reply: We would like to clarify that we used the phosphate bead as the r(0) of the Van Hove function and not the choline bead. We should also point out, that the difference between the choline and the phosphate bead r(0) is minimal, as such, we did not elaborate further on this point in the manuscript.

Referee: 2) How do authors deal with PBC in their analysis? Molecules jumping over PBC could be a serious concern. For instance in lateral diffusion studies this was emphesized as an important factor (see J. Chem. Phys. 153, 024116 (2020); https://doi.org/10.1063/5.0008312). I'm guessing that in this study sudden jumps of beads to other site of system box would contribute to error in Van Hove function.

Reply: For the 1D Van Hove function along the OZ axis, such an account was not necessary since the considered membrane molecules are located far from the edge of the system and never cross this axis. The periodic boundary conditions (PBC) for the Van Hove function in the membrane plane were as follows: The system was replicated (“cloned”) along the OX and OY axes so that the system was 9 times larger than the original one, and all beads of the original system were considered, which were now located in the center of the replicated system. It is exactly the presence of PBC that allows us to perform this operation. Thus, even if a bead from the original system would have crossed the boundary and appeared on the other side, that would not affect their distribution in the larger system.

Referee: For analysis in 2D and 3D are authors including the movement of the system? The membrane system itself is also moving - the center of mass position would differ in time. Shouldn't this be included for r(0) to avoid introduction of error due to movement of whole system?

Reply: The center of mass of the system in the case (without pressure gradients, temperatures, etc.) we presented, changes its position only due to the presence of vibrations to the atoms that make up the system.  Therefore, in the absence of macroscopic flows, additional considerations about the center mass were not needed.

Referee: I'm a little puzzled by lack of difference between DPPC and POPC systems. Since both of those lipids do not differ significantly in coarse-grain representation, is it possible that lack of difference between those two systems comes from limitation of this representation? I'll again back this up with diffusion coefficient studies, as they are also based on movement of molecules - for instance in doi.org/10.1080/08927022.2013.840902 diffusion coefficient was 4-times higher than in atomistic simulations. Are authors sure that the use of coarse-grain MD for this type of analysis does not lead to severe loss of information?

Reply: Yes, the differences between DPPC and POPC lipids are not significant in the coarse-grain representation, however, they are not identical, either. Specifically, POPC has a D2A bead, while  DPPC has a C2A bead in the same location. Importantly, the 1D VHFs intensity patterns are also not identical (see Figure 3 (b) vs Figure 3 (d)). The spatial distribution of high intensity spots (i.e., rattling dynamics of headgroup dipoles) and their time evolution in POPC and DPPC bilayers are also different, including relaxation dynamics patterns – see blue and white regions. Frequency bands are also not identical, including the height, width, and location of the peaks (see insets to Figure 4). The vibrational patterns of the membrane surface undulations of both POPC and DPPC bilayers generated by their headgroups are different (see Figure 5), including their frequency bands (see insets to Figure 6). In general, the vibrational landscapes of soft materials, including simple liquids, liquid crystals (mesogens), block copolymers, and lipid membranes are, to some extent, similar. All these systems have longitudinal and transverse phonon modes co-located. The difference is that the angle of the longitudinal modes may differ slightly indicating that the speed of sound propagates slightly different in these systems. The location of the transverse gaps in these soft matter systems is also similar and only differs slightly, implying that these systems have different diffusion properties at the picosecond time and nanometer length scales.  

Referee: I don't understand why authors are introducing frequency representation in Figure 4. It is barely discussed and doesn't seem to be relevant to the topic of the article at all.

Reply: frequency representation is important. These intrinsic frequencies are generated by the headgroup dipoles of POPC and DPPC bilayers due to their 1D lateral motions (see insets to Figure 4), and the motions along the OZ axis (see insets to  Figure 6), can be stimulated using dynamic impedance spectroscopy to study memory effects in zwitterionic bilayers (Sacci, R.L. et. al. Adv. Electron. Mater. 2022, 8, 2200121.).

Referee: Finally, are authors sure that the effect they report are due to dipole properties of lipid molecules? Lipid membranes are dynamic entity to begin with, and reported G1D such as in Fig 4 and Fig6 hardly differ from movement of any atom in the system. Fluctuations of lipid membranes, observed in time regime of tens of ns, are often used to estimate mechanical properties of lipid membranes. I think authors should mphasize more on the advantages of analyzing dipoles and the benefits of that compared to other studies. After all mphasize stress is also determined based on orientation of lipid molecules, dynamics of simulation box sizes, dynamics of area per lipid and/or membrane thickness, etc.
I also think authors should write more what additional and advantageous information does collective molecular motion provide rather than only stating that it has been less studied. Finally, they should mphasize what additional information the collective molecular motion did provide in investigated case.

Reply: Yes, this observation is supported by analytical calculations presented in the last subsection of the manuscript, entitled “Dipole-dipole interactions in lipid bilayers”. We extended the discussion of the mechanical stress and collective motions of lipids using the Van Hove function approach and incorporated this into the revised version of the manuscript.

Reviewer 2 Report

Dear authors!

The paper is interesting for me and, I suppose, for a potential reader who is specialized on membrane simulations it would be interesting also. However, I suppose that the paper is not ready for publication yet.

I picked "Major revision" to continue discussion because I am not sure that I understood it correctly. Please answer the following question.

First question. Explanation of the equations.

Could you please explain the sense of the equations in a simple form. I understand that Van Hove correlation function (let's call it VHCF) is the time-resolved "version" of g-factor (that is in turn related to radial distribution function, RDF), which is widely used to describe liquid structure. However you here talk about "dipole correlation" whereas you wrote in the equation

sumi,j(r - |ri(A) - rj(B)|)

and you further use A = Choline group and B = phosphate group. But i and j here are the indexes of any lipid molecules, aren't they? Why do you analyze correlation of choline part of one molecule with phosphate part of other one? Of course i could equal j. And in this case you have a sum of a self-term:

sumi(r - |ri(A) - ri(B)|)

and a cross-term:

sumi,j(r - |ri(A) - rj(B)|), i != j

When ri is projected to a plane or an axis, the first one is related to average rotational corellation of headgroup dipoles. The second one is related to RDF of lipid molecules, isn't it? If so, why to sum them and not consider separately as different processes? Maybe, I understood your equations improperly but it is what I've read from the text.

Second question: projection axis of 1D VHCF

I cannot understand the projection axis for 1D VHCF. You wrote about 'x-y' plane and Z-axis. I cant understand why we are talking about 'x-y' plane if the membrane is laterally isotropic. But which axis have you projected ri/j vectors? x-y: vector {1,1,0}? or Z: vector {0,0,1}? Visible features on 1D VHCF are your major and very interesting result, but how can I understand it if I cannot understand the projection axis? 

Third question: the absence of standard g-factor analysis, standard analysis of dipole rotations

As I mentioned below, VHCF is strongly related to g-factor, which is storngly related to RDF. These structure parameters should be measured for your simulations and the connections of its peaks to your peaks at VHCF should be discussed.

Author Response

Referee #2: The paper is interesting for me and, I suppose, for a potential reader who is specialized on membrane simulations it would be interesting also.

Comments:

Referee: I understand that Van Hove correlation function (let's call it VHCF) is the time-resolved "version" of g-factor (that is in turn related to radial distribution function, RDF), which is widely used to describe liquid structure.

Reply: The VHCF is not the time resolved version of g-factor (g(r), or pair distribution function, i.e., PDF), but the function of 2 variables. In contrast, g(r) is the function of 1 variable, represents only spatial correlations. Indeed, g(r) can be calculated at different times, so the time evolution of g(r) specific locations, such as the peak of the first coordination shell can be studied. It is incorrect to relate a 2-variable function (spatiotemporal correlation function) with the time evolution of a 1-variable function.

Referee: When ri is projected to a plane or an axis, the first one is related to average rotational corellation of headgroup dipoles. The second one is related to RDF of lipid molecules, isn't it?

Reply: VHCF has two parts: 1. a self-part, which describes the collective motions of particles; and 2. a distinct part, which describes the diffusion properties of particles. In this manuscript, the definitions of 3D, 2D, and 1D VHCF are most general and contain both distinct and self-parts. We clarify this point in the revised version of the manuscript.  

Referee: I cannot understand the projection axis for 1D VHCF. You wrote about 'x-y' plane and Z-axis. I cant understand why we are talking about 'x-y' plane if the membrane is laterally isotropic. But which axis have you projected ri/j vectors? x-y: vector {1,1,0}? or Z: vector {0,0,1}? Visible features on 1D VHCF are your major and very interesting result, but how can I understand it if I cannot understand the projection axis? 

Reply: We thank the reviewer for pointing this out. Indeed, the membrane is laterally isotropic. The 1D VHCF is a projection on axis OX (OY) identical along any line within the bilayer’s plane since the membrane is laterally isotropic. This definition is now clarified in the revised manuscript.

Referee: As I mentioned below, VHCF is strongly related to g-factor, which is storngly related to RDF. These structure parameters should be measured for your simulations and the connections of its peaks to your peaks at VHCF should be discussed.

Reply:  VHCF is not the time resolved version of the g-factor (g(r), or pair distribution function, i.e., PDF), but the function of 2-variables. In contrast, g(r) is the function of 1-variable. Indeed, g(r) can be calculated at different times, so the time evolution of g(r) at specific locations can be studied—e.g., the peak of the first coordination shell. However, and as mentioned previously, it is incorrect to relate a 2-variable function with the time evolution of a 1-variable function.

Importantly, in Figure 4 and 6 we represented G(r_fixed, t) functions, which are the slices of the full G(r,t) functions along their first intensity peaks at fixed “r” values.

Round 2

Reviewer 2 Report

I understood from the answer that the authors doesn't agree that VHCF can not be explained as g-factor evolution. Ok, I read it. Indeed, two-variable function cannot be fully explained by the evolution of 1-variable function. But if we average VHCF along time, is the result be g-factor? As I see from the equations, it is. So why not to provide g-factor plot alongside VHCF images to show some pictures familiar to a reader?

The section of my questions "First question. Explanation of the equations." seems to be completely missed in the answer.  But I would like to have answers to other questions of the first part.

Especially about the meaning of the sum:

sumi,jdelta(r - |ri(A) - rj(B)|)  

and why you put here phosphate from one molecule and choline from the other.

I'm sorry, but without it I cannot judge the manuscript.

Author Response

Please see Figure 4: Time evolution of the G^{1D}_{NP}(r,t) first peak height along r≈0.8 nm of lateral POPC (a) and DPPC (b) headgroup dipole spatiotemporal correlations; and Figure 6: Time evolution of the G^{1D}_{NP}(r,t) first peak height of POPC (a) and DPPC (b) headgroup dipole correlations along the z-axis. These 2D plots, see Figure 2 and Figure 3, can be sliced along either fixed "r" (time evolution of spatial correlations) or along "t" (spatial pair distribution function at fixed time). In this regard, it is redundant the discussion of g(r) function, in the context of collective dynamics of headgroup dipoles. 

We would like to clarify that we used the phosphate bead as the r(0) of the Van Hove function and not the choline bead. We should also point out, that the difference between the choline and the phosphate bead r(0) is minimal, as such, we did not elaborate further on this point in the manuscript.  We used standard definition of the Van Hove function, please, see references (44-49), including the original paper by Van Hove, reference (44).